# Evaluation of the Possible Contribution of Various Regulatory Genes to Determination of Carpel Number as a Potential Mechanism for Optimal Agricultural Yield

**DOI:** 10.3390/ijms23179723

**Published:** 2022-08-27

**Authors:** Naghmeh Abiri, Andrey Sinjushin, Dilek Tekdal, Selim Cetiner

**Affiliations:** 1Faculty of Engineering and Natural Sciences, Sabanci University, 34956 Istanbul, Turkey; 2Department of Genetics, Biological Faculty, Lomonosov Moscow State University, Leninskie Gory, 1-12, 119234 Moscow, Russia; 3Faculty of Science and Letters, Department of Biotechnology, Mersin University, 33343 Mersin, Turkey

**Keywords:** multicarpelly, gynoecium, transcription factors, miRNA, crop improvement

## Abstract

Various regulatory genes encoding transcription factors and miRNAs regulate carpel number. Multicarpelly is normally associated with increased size of the floral meristem, and several genetic factors have been discovered that influence this characteristic. A fundamental understanding of the regulatory genes affecting carpel number can facilitate strategies for agricultural yield improvement, which is crucial, given that the global population is growing rapidly. A multicarpellate plant may provide a significantly higher yield than a plant bearing fewer carpels. Higher yields can be achieved via various means; in this review, we provide an overview of the current knowledge of the various regulatory factors that contribute to multicarpelly and the potential of increasing carpel number to achieve an increased yield.

## 1. Introduction

The Green Revolution of the 1960s dramatically improved the yields of major staple crops (wheat, corn, and rice) to address the needs of an increasing global population and provide food security. By the 1980s, transformation technologies provided engineered traits with clear benefits to developing countries and farmers. However, the acceptance of genetically modified traits is controversial in some countries, where the cultivation of genetically modified plants is banned [1]. To date, only 28 countries allow the cultivation of GM crops, whereas others permit their import for food and feed use.

Recent molecular breeding and genomic technologies have sped up breeding and trait development to improve productivity and environmental flexibility. The success of future food security demands the recruitment of traits that are not found in wild-growing crop precursors.

The evolution of domesticated forms of plants involved the selection of traits that were suited to humans’ needs rather than the wild environment. Plants cultivated for these traits are worth the labor of cultivation. The change from a wild plant to a crop requires substantial morphological and physiological adaptation. Crops with similar uses display similar trait changes, coined the domestication syndrome. The domestication syndrome can be defined as the characteristic set of phenotypic traits associated with the genetic change to a domesticated form of an organism from a wild ancestral form [2,3].

The Fabaceae (or Leguminosae, commonly known as the legume, pea, or bean family) family accounts for the majority of the world’s main food crops. Comprehension of the regulatory mechanisms underlying the multicarpellate trait can be helpful to improve the yield of legume crops. Multicarpelly has considerable potential to increase yield [4,5,6], which we will further discuss. The yield per multicarpellate plant may be significantly higher than that of a plant with its flowers bearing one or two carpels. 

Although there are several ways to obtain an increased yield, in this review, we focus on the genes and their products contributing to multicarpelly, as well as the potential of increasing carpel number to achieve increased seed yield.

## 2. Natural Variation in Gynoecium Structure

Although ovules completely enclosed in a carpel are a distinguishing feature of angiosperms, the exact origin and nature of the carpel remain a matter of debate [7]. There is no room to discuss all existing controversies in a given paper, so we will hereafter refer to a carpel as an entity without further exploration of its possible evolutionary origin.

In the case of the whorled arrangement of floral organs, each floral whorl can be characterized with respect to its merism (or merosity), i.e., the number of organs per whorl or whole floral domain, such as the gynoecium. Accordingly, the gynoecium can be monomerous (if it consists of a single carpel, which can hardly be interpreted as a whorled), di-, trimerous, etc. A gynoecium consisting of several carpels is referred to as apocarpous (free carpels) or coenocarpous (or syncarpous if carpels somehow fuse; Figure 1D,F). In many angiosperm groups, the gynoecium merism is more or less uniform among all representatives. For example, all genera of the Brassicaceae possess dimerous gynoecia. In other taxa, the gynoecium merism may vary in a relatively wide range, even when induced mutants or cultivars are not considered. Within a single genus, *Paris* (Melanthiaceae), gynoecium merism ranges between four (rarely three) in *P. quadrifolia* and 10–12 (*P. japonica*). The number of carpels among representatives of the tribe Potentilleae (Rosaceae) varies from one (*Alchemilla*) to several hundred (*Fragaria*), although in a spirally arranged rather than whorled gynoecium.

During the course of evolution, the amount of seeds produced by a single flower may either decrease or increase. One way to alter seed productivity is to develop more/fewer ovules per single carpel. As evidenced by studies of model plant species, the number of ovules is controlled by multiple genes [8]. Some of these genes simultaneously control the size of the gynoecium, thus allowing for initiation of more/fewer ovules, although these two parameters are not correlated as strongly as might be expected [9].

The other way to change productivity, which is more related to the scope of the present review, is to modify the number of carpels per flower, i.e., the gynoecium merism. In many cases, a reduction in carpel number results in the production of the so-called pseudomonomerous rather than true monomerous gynoecia, i.e., only one functional carpel persists, but additional carpel(s) (more or less rudimentary) are still traceable. Some Anacardiaceae have a tricarpellate but pseudomonomerous gynoecium [10]. In some particular cases, the number of (functional) ovules may decrease relative to that of carpels. In the dimerous gynoecium of *Trapa* (Lythraceae), each of two carpels contains one ovule, although typically, only one seed matures. In addition to true monomerous and pseudomonomerous types, Sokoloff et al. [7] recognized a special gynoecial pattern, viz., mixomerous. In such a case, several carpels unite, producing a unilocular ovary, and lose their individuality, so it is impossible to distinguish to which of the carpels the ovule(s) belong(s) [7]. Mixomery is often associated with central placentation and reduction in ovule number. For example, in *Juglans* (Juglandaceae), a single ovule is shared between two equally developed carpels, whereas in *Fagopyrum* (and some other members of Polygonaceae), only one ovule is enclosed by three fused carpels. In such a case, several carpels unite to produce a unilocular ovary and lose their individuality, so it is impossible to distinguish from which of the ovule(s) the carpels originated. 

Carpel numbers can vary within and between species. Most angiosperms have gynoecia with two to five carpels, whereas gynoecia with higher merism can be referred to as multicarpellates [11]. The gynoecium of most angiosperms consists of more than one carpel, most of which (more than 80%) are fused (congenital fusion). Approximately 10% of species have free carpels (apocarpy), and the remaining 10% display a monocarpellate gynoecium (monomerous) [12]. As Endress (2014) concluded, multicarpelly is usually found in polysymmetric flowers lacking synorganization in/between the androecium and gynoecium. Additionally, polymerous flowers are more prone to stochastic fluctuations in their merism than oligomerous flowers [13]. This statement is not restricted to gynoecia with a merism exceeding five but can be expanded to all cases in which the gynoecium of a particular genus/species has an unusually high merism compared with related taxa. Several cases are briefly described hereafter in which genera/species with multicarpellate flowers are nested within clades characterized by reduced gynoecial merism.

Most plants with the monomerous gynoecia (more than 19,000 species) are found in the legume family (Figure 1A). Within the legume family, there are only rare records of multicarpellate flowers. Species with multicarpellate gynoecia exist in approximately 15 genera. The ancestral legume gynoecium is monomerous, and multicarpelly evolved at least seven times independently, although it is not a common condition in the family [14].

One of the leguminous species that regularly produces multicarpellate gynoecia is *Vuralia turcica*. This plant (previously named *Thermopsis turcica*) is endemic in Turkey and locally known as piyan. The existence of a prominent trait, a gynoecium with two to four fully developed carpels, makes this plant different from other species of the Papilionoideae subfamily [15]. This species is distributed in Konya and Afyonkarahisar around Eber and Akşehir lakes in Turkey. It has been listed as a critically endangered (CR) species in the “*Red Data Book of Turkish Plants*” [16,17]. The most significant characteristic of floral morphology in *V. turcica* is its gynoecium. It is almost always composed of three carpels, which are of the same size and seem to develop somewhat asynchronously (Figure 1C). Normally, all three carpels develop to mature pods. In some rare cases, the gynoecium consists of one, two, or four carpels (with a low frequency of 2–3% for one and two carpels and 0.13% for four carpels) [18]. 

*V. turcica* combinates an unusual tricarpellate gynoecium with a monosymmetric ‘flag blossom’ [18]. However, as distinct from many other unicarpellate papilionoid legumes, this plant has completely free (unfused) stamens and keel petals connected only with tips, i.e., a relatively moderately expressed floral monosymmetry. All other known polymerous gynoecia in Leguminosae are found in flowers with either polysymmetry (*Inga* p.p., *Acacia* p.p.) or with a very special kind of monosymmetry not involving an orchestrated interaction between different floral domains (*Swartzia* p.p.) [18] (and papers cited therein). More examples of legumes with multicarpellate gynoecia are known [14,19] and often reported as anomalous forms of already known species [20,21].

*Tupidanthus* (Araliaceae), with up to 200 carpels in its gynoecium, finds its closest affinities in the so-called Asian Schefflera clade [22] with mostly pentamerous gynoecia. A multicarpellate state arose independently in several lineages of Araliaceae [23]. Flowers of *Decumaria* (Hydrangeaceae) are remarkable with their high merism in all floral whorls, including the gynoecium, usually hexamerous and irregularly shaped, whereas most other representatives of Hydrangeaceae have di-, tri, or pentamerous gynoecia [24]. More examples are listed by Endress [11].

Flowers with extremely multicarpellate gynoecia [24,25] are reminiscent of fasciated mutants of model plant species (e.g., *clavata* of *Arabidopsis thaliana*, Brassicaceae; see below). We define fasciation as a set of morphological anomalies resulting from an atypical increase in meristem, either apical or floral. Such an increase may be due to the fusion of several normally independent meristems, which is usually not heritable and may appear in any taxon. Alternatively, fasciation may arise when negative control of meristem size is lost, usually because of mutation in a certain gene(s). This aspect will be discussed in more detail in the succeeding sections.

One of the features of similarity between multicarpellate flowers of wild-growing taxa and fasciated mutants is the unusual shape of the receptacle and especially of the gynoecium closure [11]. Instead of a regular whorl, multiple carpels follow an anomalous contour, sometimes elliptic or even H- or asterisk-shaped (*Tupidanthus*: [25]). As suggested by Choob and Sinyushin [26], the spatial patterning of an anomalously enlarged (fasciated) apical or floral meristem needs to provide the optimal density of vascular bundles in the tissue volume. For these purposes, such a meristem usually becomes elliptic, flattened ridge-like or asterisk-shaped (monstrose) or even splits into several smaller meristems (defasciation). All these transformations can be found in both fasciated shoots [26] and flowers, such as those of *Tupidanthus* (see Figures 3, 5 and 59 in [25]) and other genera [11].

In some multicarpellate flowers (e.g., *Gyrostemon*, Gyrostemonaceae), a central part of FM remains uninvolved in morphogenesis and undifferentiated (see Figures 20–25 in [11]).

More carpels may initiate if FM continues proliferation for a longer time, as is likely the case in some Ranunculaceae (*Myosurus*) and Rosaceae (*Fragaria*, *Potentilla*). This pattern can also be exemplified by flowers of the fasciated mutants of *A. thaliana* (such as *clavata1*) producing additional whorls of carpels inside the already initiated gynoecium [27]. In *V. turcica*, both increase in FM sizes and its prolonged proliferation probably contribute to the origin of gynoecium polymery [18]. When carpel number is reduced or increased via shifts in FM sizes or proliferation activity, such changes would hardly leave any intermediate stages or rudiments, as opposed to the cases of pseudomonomery.

The number of floral organs can increase through a secondary proliferation/splitting of a few primarily initiated primordia. This pattern is a relatively common mode of polymerization of the androecium [28] but not of the gynoecium. Such multiplication is probably responsible for multicarpelly in *Nolana* (Solanaceae), although this case deserves a special revision [11].

The morphology of gynoecium and the resulting fruit is such a remarkable feature, that many taxa received their generic status mostly because of their multicarpelly. Segregation of tricarpellate *V. turcica* into a monotypic genus, *Vuralia*, found no support [29]. Similarly, *Tupidanthus* is recognized as a member of the Asian group of *Schefflera* [22], whereas *Decumaria* is most appropriately included into *Hydrangea* [30]. As demonstrated by these examples and especially by the existing ideas on the regulation of FM sizes (see next sections), changes in merism may occur easily, so there are recurrent (but unsupported by molecular phylogenetic studies) temptations to overestimate the taxonomic significance of recent morphological novelties.

The dissection of molecular mechanisms underlying the increase in both ovule and carpel number is important not only in terms of understanding principles of angiosperm evolution but also for crop improvement. Various modes of increase in carpel number are schematically exemplified in Figure 1 and will be discussed in the following sections.

## 3. The Structural Organization in Floral Meristem

Plants have two growth phases during their life cycles, viz., vegetative and generative. The aboveground part of the plant organism develops from the apical meristem. This vegetative meristem gives rise to all of the leaves that are found on the plant. Meristem identity changes during plant development. The plant maintains its vegetative growth until the apical meristem undergoes a change. After the vegetative phase, plants initiate the floral transition in response to both environmental and endogenous cues to optimize reproductive success. 

The shoot apical meristem (SAM) is often subdivided into zones differing in their properties and functions [26]. SAM possesses a stem cell population at its apex that is defined as the central zone (CZ). Within the CZ, stem cells slowly divide, and the daughter cells are translocated into the sidelong flanks of the CZ, forming the peripheral zone (PZ) [31]. Cells in the PZ later develop into lateral organs, such as leaves and flowers. There is another functional zone called the rib meristem (RM) that is located underneath the central zone and helps to specify the identity of stem cells in the CZ. There is also the organizing center (OC), a group of cells located at the center of the RM [31]. In the case of simple racemose inflorescence (such *A. thaliana*), SAM, which generates leaves and branches, turns into an inflorescence meristem after the floral transition. Therefore, SAM obtains inflorescence meristem (IFM) identity. The IFM gives rise to FM at its flanks in a spiral phyllotaxy. The inflorescence meristem often produces modified leaves (bracts, sometimes cryptic) bearing floral meristems (FMs) in their axils, whereas FMs develop into flowers. Unlike SAM, which is usually active throughout the entire plant lifespan, FM activity is inhibited by perfectly synchronized developmental programs that assure the exact formation of floral organs [32,33]. 

When flowering begins, it causes a new developmental program that results in the formation of reproductive organs. The transition to reproductive development is accompanied by several changes in the physiology of the plant [34].

Furthermore, mutual interactions between members of the floral meristem identity class of genes is a key feature that allows for a sharp transition between vegetative and reproductive development and guarantees that no reversion will occur once the decision to flower has been taken, thereby providing robustness to the gene regulatory network [34]. However, if no terminal flower is produced, the IFM may revert to its vegetative state, as is the case in various taxa, such as *Chamaecytisus* spp. (Leguminosae). These perennial plants may produce terminal compound inflorescences, but during the next season, the same IFM continues proliferation as vegetative shoot.

FM is subject to a series of developmental changes that eventually give rise to the four basic structures of the flower, including sepals, petals, stamens, and carpels. Generally, four whorls develop from the floral meristem, and each of the flower structures is derived sequentially from a whorl. The outer whorl is the first to appear, and it develops into the sepals. The second whorl develops into petals. The third and fourth whorls define the stamen (male reproductive organs) and carpel (female reproductive organs), respectively. This is the case of *A. thaliana*, whereas in other taxa, the order of initiation of floral primordia may be diverse [35].

## 4. Genes Defining Identity of Floral Organs and Their Possible Contribution to Multicarpelly

Genetic factors contributing to the number of carpels produced by flowers are summarized in Figure 2.

From a genetic perspective, two phenotypic changes are programmed in the plant that control vegetative and floral growth. The formation of flowers depends on the proper functioning of these genetic changes. Based on the development of plant organs in a sequential manner, it has been suggested that a genetic mechanism exists whereby a series of genes are sequentially turned on and off. This switching is necessary for each whorl to acquire its final unique specification.

The ABC model was proposed in the 1990s to explain the development of floral organs [36]. This model is based on floral homeotic mutants of *Arabidopsis thaliana* [37] and *Antirrhinum majus*, proposing that the combinatorial action of three sets of genes, the A-, B-, and C-function genes, specifies the four floral organs (sepals, petals, stamens, and carpels) (Figure 2). Sepals are specified by A-class genes, including *APETALA1* (*AP1*) and *APETALA2* (*AP2*). According to this model, petals are specified by A- and B-class genes. B-class genes include *APETALA3* (*AP3*) and *PISTILLATA* (*PI*). Stamens are specified by B- and C-class genes, such as *AGAMOUS* (*AG*), and carpels are specified by C-class genes [37,38]. If B-related function is impaired, stamens transform into carpels, and the corolla becomes sepaloid. Similarly, simultaneous loss of function of A- and B-class genes produces carpel-like structures in all floral whorls [39,40,41]. Hence, extra carpels can be differentiated from primordia normally destined for other purposes rather than initiated de novo.

Later, an extension of the ABC model was introduced, proposing that all the identities of floral organs are specified by A, B, C, D, and E floral homeotic genes [42,43]. According to this model, floral organ formation occurs by different combinations of these four gene functions. Molecular cloning of the initial ABC model genes revealed that *AP1*, *PI*, *AP3*, and *AG* encode members of the MADS family of transcription factors. The ABC model suggests that different combinations of MADS proteins activate different groups of target genes in each whorl. How homologous MADS transcription factors are modulated to obtain whorl-specific functions at the molecular level remains unknown, especially in non-model species.

Some homeotic changes may have a certain evolutionary significance. The expansion of B-class gene expression to the outer perianth whorl seems to result in petaloid tepals in many monocots [44]. Some of the naturally occurring floral homeotic mutants possess sufficient fitness to form stable populations [45]. Although some studies revealed that the formation of the multicarpellate trait may be the result of stamen-to-carpel homeosis [19,42,43], this pattern scarcely contributes to the evolutionary increase in carpel numbers.

## 5. Key Players in Regulation of FM Size: *CLV/WUS* Negative Feedback Loop

It has been established in detail which mechanisms govern the spatial arrangement of phyllomes (phyllotaxis) in both vegetative stems and flowers. As indicated by a series of experiments, each emerging primordium becomes a site of polar auxin transport through the meristem and therefore acquires a surrounding inhibitory field that prevents the inception of the next primordium [46]. The resulting morphology is therefore dependent on the ratio between sizes of stem/flower meristem and initiating primordia. If areas of the inhibitory field of each primordium are constant, more floral organs per whorl can initiate as the floral meristem (FM) is enlarged. The merism of each floral whorl (or domain) can result either from FM expansion (permitting inception of more organs per whorl) or its longer proliferation (supporting the initiation of additional whorls). This section is dedicated to the regulation of FM size, whereas in the next section, we focus on the control of the time of FM proliferation.

A homeodomain transcription factor, *WUSCHEL* (*WUS*), positively regulates meristem development. During postembryonic development, *WUS* is crucial for maintaining the stem cell population [47]. *WUS* transcripts are specifically expressed in OC. The *CLAVATA* (*CLV*) genes (including *CLV1*, *CLV2*, and *CLV3*), on the other hand, negatively regulate the size of the shoot-stem cell population by confining *WUS* activity to a limited group of OC cells. Mutations in the *CLV* genes result in the overproliferation of SAMs and floral meristems [48] (Figure 1G). 

There are two known regulatory pathways governing FM size and proliferation in *Arabidopsis*. One of them includes *WUS* and *CLV*, and the other involves *AG* and *WUS*. *WUS* and *CLV3* form a negative feedback loop that promotes cell–cell communication between the CZ and the RM to maintain the appropriate SAM volume. The *CLV3* signal inhibits *WUS* expression and restricts the *WUS* transcript to the OC, whereas *WUS* positively regulates the expression of *CLV3* in the CZ [31]. After the initiation of carpels, the floral meristem ceases to maintain a stem cell population (called determinacy). The *CLV/WUS* pathway components are implicated in the development of multilocular fruit trait in cultivated mustard varieties (*Brassica*); some examples will be provided in the last section.

When negative control of SAM and/or FM is lost, the resulting phenotype is usually referred to as fasciation. It is obvious that some regulatory patterns are shared between SAM and FM (such as *CLV*/*WUS* pathways), and mutants at involved loci exhibit both stem and flower fasciation. However, some pathways may be associated with only one type of meristem. For example, homozygotes at mutant alleles of genes *FASCIATA1* and *FASCIATA2* (*fas1*, *fas2*) of *A. thaliana* exhibit pronounced stem fasciation but fewer perianth organs compared with wild-type plants, with unaffected carpel number [49].

## 6. FM Should Not Proliferate for Too Long: *AG/WUS* Interaction

There is a coincidence between the termination of floral stem cell fate and the development of carpel primordia, i.e., the last (topologically but not often temporarily) organs to be produced from the FM. However, the termination of floral stem cell maintenance does not simply comprise the differentiation of FM cells into carpel cells. An independent mechanism that is coordinated with organ identity specification is responsible for the precise termination of the FM [50]. 

The timing of FM termination is precisely controlled by another feedback loop (*AG/WUS*). *AG*, a floral homeotic MADS-box gene, is a key component of this regulatory pattern. In the flower developmental stage, *WUS* and the floral meristem regulator, *LEAFY*, induce *AG* expression in whorls 3 and 4 of the floral primordia, where stamens and carpels will later form. In *A. thaliana*, *AG* terminates floral stem cell maintenance by repressing *WUS* expression. This repression of *WUS* expression is either direct or indirect. Sun et al. [51] showed that *AG* represses *WUS* expression indirectly by activating *KNUCKLES* (*KNU*), which in turn represses *WUS* expression directly or indirectly. *WUS* expression is shut off about two days after the induction of *AG* in an *AG*-dependent manner, when carpel primordia are specified. In later stages, *AG* is continuously expressed in developing stamens and carpels to regulate reproductive development [51]. Furthermore, *KNU* was identified as a key link in the feedback regulation of *WUS* by *AG*. *AG* directly regulates *KNU*, a potential repressor of *WUS* transcription. Overexpression of *AG* does not terminate the floral meristem prematurely, but the overexpression of *KNU* is sufficient to cease *WUS* expression and terminate the stem cell population before due time. When *KNU* is mutated, the expression of *WUS* is prolonged, resulting in an increased number of carpels and stamens (the potential role of *KNU* in multicarpelly will be discussed further) [51].

On the other hand, Liu et al. [50] revealed that *AG* can also directly repress *WUS* expression by recruiting Polycomb group (*PcG*) genes to *WUS*. Genetic studies support the notion that *KNU*- and *PcG*-mediated pathways act downstream of *AG* and in parallel to each other to terminate floral stem cell maintenance [50].

The *KNU* locus and its contribution to flower and carpel development were identified in 2004 by Payne et al. [52], who concluded that *KNU* encodes a small protein, including a single C2H2 zinc finger and containing an EAR-like active repression domain, which probably functions as a transcriptional repressor. One of the largest TF families in plants is zinc finger proteins (ZFPs), which are categorized into subfamilies according to the order of the Cys and His residues in their secondary structures [53]. C2H2-type zinc finger protein genes encode proteins that play many roles in plant growth, development, and biotic stress resistance. C2H2-ZFPs are extensively involved in regulating flowering induction and floral organ development. According to Payne et al. [52], *KNU* expression occurs early in the development of the gynoecium and persists near its base until after ovule primordia appear. The structures localized at the gynoecium base, such as the gynophore and nectaries, are suppressed by *KNU*, allowing the ovaries to develop fully. *KNU* is expressed most strongly in flower buds. In all cases, *KNU* transcripts have been detected in a wide range of cell types comprising the floral organs [52].

*KNU* belongs to the *SUPERMAN-like* family, members of which encode C2H2-ZFPs involved in the suppression of FM size [54] (and references cited therein). Gene *SUPERMAN* (*SUP*) is a negative repressor of *WUS*. In grapes (*Vitis vinifera*, Vitaceae), multicarpellate berries develop in plants with reduced expression of *VvSUP-like* [55]. It was confirmed that *VvSUP-like* prevents the expression of grape orthologues of *SUP* targets *VvAG1* and *VvWUS* [55]. As tri- and tetracarpellate berries are larger, this feature is of agricultural value.

In another study, Shang et al. [56] demonstrated that KNU protein directly binds to the *CLV1* locus, as well as the cis-regulatory element on the *CLV3* promoter, and represses their expression during FM determinacy control. KNU also physically interacts with WUS, and this interaction prevents *WUS* from maintaining *CLV3* in the central zone. Shang et al. reported a regulatory framework in which KNU plays a position-specific multifunctional role in FM determinacy [56]. The authors proposed that KNU is involved in the regulation of FM size and observed that the floral buds of *knu* mutants have a larger FM size, which may correspond to more carpels on the flowers [56]. 

Increased expression of *WUS* in *Arabidopsis* leads to an increase in the number of floral organs, specifically locules [57]. In addition, repression of *WUS* in tomato results in a reduction in locule number [58]. Bollier et al. [59] studied *A. thaliana* Mini Zinc Finger2 (*AtMIF2*) and its orthologue in tomato (*Solanum lycopersicum*), Inhibitor of Meristem Activity (*SlIMA*). *AtMIF2* and *SlIMA* engage with their respective *KNU* genes to form a transcriptional repressor complex. *AtMIF2* and *SlIMA* are bound to the *WUS* orthologues in the respective plants, inhibiting their expression. The results of this study suggest that *AtMIF2/SlIMA* is involved in determining locule number through the regulation of *WUS* expression; additionally, the authors emphasize the importance of *AtMIF2/SlIMA* during the determination of carpel number and therefore fruit size [59].

## 7. Further Regulatory Genes Potentially Contributing to Multicarpelly

The *SUPERMAN* (*SUP*) gene encodes a TF with a C2H2-zinc finger DNA-binding domain and an EAR repression domain. During flower development in *Arabidopsis*, *SUP* controls cell proliferation in the stamens and carpel primordia, as well as the ovules. *SUP* functions as a boundary gene to separate the stamen produced by whorl 3 from the carpel produced by whorl 4 [60]. The boundary-specific function of *SUP* is to repress the expression of B genes (*AP3* and *PI*) by modulating auxin- and cytokinin (CK)-regulated processes [61]. Allelic series analysis of *SUP* showed that late *SUP* functions play a role in organ morphogenesis by controlling intra-whorl organ separation and carpel medial region formation. The *sup* (epi)alleles and *sup-5* showed an excess of carpels. All the flowers of *sup-5* are bisexual and exhibit whorl 3 and 4 indeterminacy, i.e., more stamens and carpels (average of 3) than in the wild type [61]. 

In a study conducted on *A. thaliana* by Xu et al. [62], *sup* mutants showed reduced sizes of floral organs and increased carpel numbers. These results suggest that the function of *SUP* could be associated with organ differentiation. The authors also showed that polar auxin transport contributes to the formation of extra stamens and carpels in *sup* mutants [62]. The study revealed how SUP functions control floral organogenesis and FM size non-cell autonomously. *SUP* negatively regulates the expression of the auxin biosynthesis genes (*YUC1*/*4*) in the stamen-to-carpel boundary region. In *sup* mutants, the repression of *YUC1* and *YUC4* genes elevates auxin levels at the boundary between whorls 3 and 4, which leads to an increase in the number and the prolonged maintenance of floral stem cells, as well as the consequent production of extra reproductive organs [62]. 

The *JAGGED* (*JAG*) gene is a key regulator of lateral organ development which encodes a putative C2H2 zinc finger transcription factor. Although *JAG* expression is identified in all floral organs, mutation of *jag* alleles strongly affects sepal and petal development, suggesting that *JAG* may act redundantly with other factors in stamens and carpels. *NUBBIN* (*NUB*), a paralogue gene of *JAG*, is responsible for this redundancy. *NUB* acts redundantly with *JAG* to promote the differentiation of carpels and stamens. Researchers have proposed that *JAG* can stimulate organ growth by promoting cell proliferation. *JAG* is also a direct target gene of the C-class gene *AG*. In contrast to *JAG*, *NUB* is particularly expressed in leaves, stamens, and carpels [54,63].

The rice *STAMENLESS 1* (*SL1*) gene is orthologous to the *Arabidopsis* gene *JAG*. Loss of function of *SL1* (in *sl1* mutant) inhibits the expression of B-class genes in whorls 2 and 3, which results in homeotic conversions of the two whorls into palea/lemma-like structures and carpels, respectively (Figure 1J). *sl1* flowers exhibited palea/lemma-shaped lodicules elongated to various extents. In addition, stamens were changed into organs with variable expression of carpel features. In other words, the *sl1* mutant produces varied numbers of carpels [64].

The *A. thaliana* genome contains five class III homeodomain-leucine zipper genes. Among them, the *PHABULOSA* (*PHB*)*, PHAVOLUTA* (*PHV*), and *CORONA* (*CNA*) genes were shown by Prigge et al. [65] to have the greatest effect on carpel number. The *phb*, *phv*, and *cna* mutants form extra carpels. The inflorescences of *phb cna* double mutants displayed short internode lengths and occasionally produced flowers containing extra carpels [65].

Among other mutants of *A. thaliana* exhibiting stem fasciation, *mgoun1* and *mgoun2* (*mgo1*, *mgo2*) are remarkable, with a SAM split into several proliferating apices and a slight increase in carpel number [66]. Gene *MGO1* encodes a topoisomerase most possibly involved in *WUS*-mediated control of SAM size through chromatin remodeling. The exact mode of interaction between *MGO1* and *WUS* is obscure, but they seem to act synergistically without direct interaction [67].

MicroRNAs (miRNAs) are 21–22-nucleotide-long noncoding RNAs. They have been considered extensively and determined in various plant species [68,69,70,71,72]. miRNAs play an inseparable role in flowering due to their function in post-transcriptional genetic regulation. Individual miRNAs display unique sequences; however, they are highly conserved across species within the plant kingdom. miRNAs are involved in almost all aspects of plant growth and development. The modification of plant miRNAs provides the opportunity to enhance economically important functions, including increased crop yield, plant development, abiotic and biotic stress, and flowering time. Consequently, the genesis of superior cultivars can contribute to agricultural and horticultural production. Several studies have indicated that most of the essential miRNAs for biological processes target TFs or stress-response factors. In the absence of miRNAs and their regulation, plants would suffer developmental deficiencies in many critical developmental stages [73,74]. 

It was shown that overexpression of miR156 in tomato plants resulted in the formation of extremely modified ovaries composed of multiple fused extra carpels and undifferentiated tissue inside the post-anthetic ovaries [75]. miR156 targets several *SQUAMOSA PROMOTER BINDING PROTEIN-LIKE* (*SBP*/*SPL*) family genes. SPL proteins are a diverse family of TFs that play an important role in plant flower and fruit development, architecture, and growth. A highly conserved region of 76 amino acids called the SBP domain characterizes SPL proteins [76,77,78,79]. Overexpression of miR156 in tomato led to downregulation of most *S. lycopersicum SBP*-box (*SlySBP*) genes (15 *SBP/SPL* family members identified in tomato) in developing ovaries and to the formation of fruits containing ectopic organs, such as extra carpels and leaf-like structures [75].

In another study, *Arabidopsis squint* mutants (*sqn*) displayed an increased expression level of miR156-targeted *SPL*s and developed siliquae with an increased number of locules [80]. 

Shang et al. [56] generated artificial microRNAs of *KNU* driven by *CLV3* and *WUS* promoters in order to reduce *KNU* activity in the CZ and OC, respectively. Flowers of both *pCLV3:amiR-KNU* and *pWUS:amiR-KNU* transgenic plants of *A. thaliana* produced three to four carpels compared to the wild type having two carpels. The result indicated that compromised *KNU* expression in either CZ or OC leads to enhanced FM activity and evidenced the role of *KNU*, *WUS*, and *CLV3* in the possible origin of multicarpelly [56].

## 8. Potential Use of Certain Mutations in Agriculture

A single-nucleotide mutation in a *CLV3* gene orthologue of *Brassica rapa* results in an *ml4* mutant exhibiting multilocular trait [4]. The silique of *B. rapa* normally consists of two locules. Multilocular (more than two locules) lines of *B. rapa* (variety Sangribai) have been discovered in nature [81]. The *ml4* mutant shows normal inflorescences and siliquae but increased carpel numbers (3.9 carpels on average), in addition to exhibiting an extra gynoecium inside the silique with high frequency. The formation of the multilocular phenotype is simultaneous with enlarged SAMs, which could cause an enlarged FM and therefore enable the initiation of more floral organ primordia than in the wild type [4]. Such a floral phenotype is strongly reminiscent of fasciated mutants of *A. thaliana*, such as *clv1, clv2*, or *clv3* (see above; Figure 1G).

Fan et al. [4] reported the isolation and functional characterization of the *CLV3*-like gene in *B. rapa*. A novel single-nucleotide mutation (C-to-T) in *BrCLV3* is essential for the control of SAM size and locule and seed number per silique. This point mutation led to an amino acid change of Pro-to-Leu (*ml4* mutant), resulting in the formation of the multilocular trait in *B. rapa*. Expression analysis carried out in this study showed that the putative negative pathway in the feedback loop involving *CLV3* and *WUS* was disrupted in the *ml4* mutant [4].

Similarly, multilocular phenotypes detected in the *Brassica juncea* cv. Duoshi are caused by mutations in *BjLn1*, a *CLV1* homolog. Most of the germplasm resources in *B. juncea* contain siliquae with only two locules, whereas a few varieties, such as Duoshi, Santong, and Sileng, produce siliquae with three or four locules. It has been demonstrated that two genes (*Bjln1* and *Bjln2*) are responsible for multilocular siliquae in *B. juncea* cv. Duoshi [81]. In another study, the *BjLn1* gene was successfully isolated through the map-based cloning method. Xiao et al. [5] showed that *BjLn1* could rescue the multilocular phenotype. Multilocular fruit habit in *B. juncea* is associated with two missense mutations and large deletion in *BjCLV1*. Mutations reduced *Bjln1* function and expression level, which interrupted the *CLV/WUS* signal pathway, leading to the enlargement of the shoot and floral meristem and the formation of the multilocular siliquae [5] (Figure 1G). 

Another mutant of *B. juncea* showed a trilocular phenotype caused by an insertion into the coding region of a second *CLV1* homolog, *BjMc1*. Using fine mapping and transgenic complementation, Xu et al. [82] cloned the *BjMc1* gene controlling carpel development in *B. juncea*. An insertion of a copia-LTR retrotransposable element (RTE1) disrupts the transcription of *BjMc1* so that the *BjMc1* gene can encode a dominant negative protein, causing the trilocular silique phenotype [82].

In several other species of *Brassica*, including *B. napus*, the multi-silique trait was recorded. In such plants, one flower occasionally produces several bilocular (i.e., similar to normal) siliquae rather than a single multilocular silique [83] (Figure 1H). The exact molecular mechanism of such transformation remains unclear, although such a habit is potentially valuable for agriculture.

In tomato, an increase in the number of fruit locules was reported when the homologues of both *CLV1* and *CLV3* were mutated [58]. The wild ancestor of tomato had a small bilocular fruit (Figure 1D), whereas gynoecia of modern varieties contain more locules. The fruit of the domesticated tomato is larger compared to its wild type. During domestication, both carpel cell division and carpel number determine the final size of tomato fruit [84]. The *fasciated* (*fas*) and *locule number* (*lc*) mutations affect the number of fruit locules (Figure 1E). Most cultivated tomato varieties possess mutations in either the *fas* or both genes. The multilocular *fas* phenotype results from a mutation in the regulatory region of a *CLV3*-related gene, whereas the *lc* trait is caused by 2 SNPs in a repressor element downstream of a *WUS* gene homologue [58].

A floral organ mutant was observed in transgenic *Medicago truncatula* with two pistils (*bi-pistil*, *bip*) [85]. A *bip* mutant is remarkable, as it is probably the only known legume mutant that produces more than one fertile pod per flower. For comparison, *stamina pistilloida* mutants of garden pea (*Pisum sativum*) have two or more stamens converted into extra carpels (Figure 1B), which usually remain underdeveloped and do not contribute to yield [19].

Some of the cultivars of sweet cherry (*Prunus avium*, Rosaceae) develop twin fruits, especially under heat stress, suggesting a heritable nature of this anomaly. This habit is associated with several MADS-box genes, especially *PaMADS12* and *PaAG*, a sweet cherry orthologue of *AG* [86]. These genes encode TFs and exhibit a temperature-dependent expression pattern, again confirming the role of the *AG*-dependent pathway in the control of carpel number [86].

There are several other crop species in which productivity is increased through flower fasciation, resulting in larger or twin fruits. Among these are kiwifruit (*Actinidia deliciosa*, Actinidiaceae: [87]) and strawberry (*Fragaria* × *ananassa*, Rosaceae: [88]). In the latter case, the authors recorded that fasciation was accompanied by reduced expression of *FaTFL1*, a strawberry orthologue of *TERMINAL FLOWER1* [88]. This association is unusual but hardly causative, as the *TFL1* gene, initially discovered in *A. thaliana*, regulates flowering time and determinacy of SAM rather than sizes of SAM and FM [89] (and papers cited therein).

## 9. Conclusions

Studies on model plants reveal that there are several possible ways to increase the number of carpels produced by a single flower. Although found in developmental mutants, stamen-to-carpel homeosis does not play a significant role in either floral evolution or breeding. A deeper examination of non-model objects, including valuable crop species, indicates that in most cases, an increased carpel number (and accordingly, larger fruits and higher yield) is associated with weakening or loss of negative control of floral meristem size. This control primarily relies on the *WUS* gene, a key marker of stem cell properties in plants, which is subject to negative regulation by numerous genes.

A considerable portion of plant-derived products is either a direct derivative of the carpel or a byproduct of a flowering plant. Even when the consumed product is not seeds or fruits, the effective propagation of many agriculturally valuable plants suggests the acquisition of seeds. Enhancing seed yield through the genetic engineering of crops has long been a primary goal of agricultural research. Therefore, increasing the yield per plant in order to continue to benefit from high yields is required for the cultivation of new varieties. An improved understanding of multicarpelly-related genes orchestrating gynoecium development in various species has the potential to provide novel strategies for crop improvement. Thus, a closer look at plants such as *Vuralia turcica* with a unique characteristic of a gynoecium with 2–4 fully developed carpels at genomic, transcriptomic, and epigenetic levels may shed light on the matter.

A millennia-old process of breeding, both intuitive and science-driven, repeatedly and independently produced forms with increased carpel number. This feature may be considered a component of domestication syndrome in certain species. Ongoing breeding activity may recruit already known mutations affecting carpel number or produce forms with a desired floral morphology using genome editing or other gene-directed means.

## Figures and Tables

**Figure 1 ijms-23-09723-f001:**
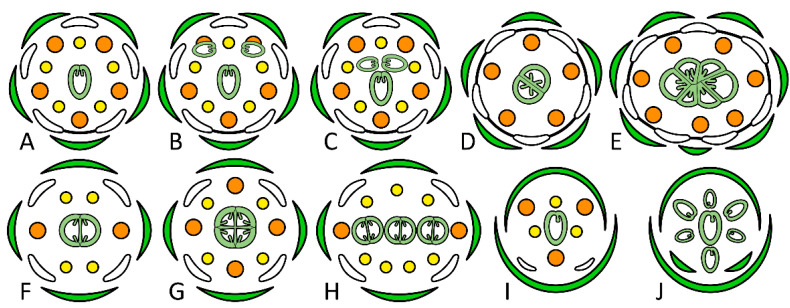
Floral diagrams illustrating normal and anomalous multicarpellate morphologies. (**A**) typical flower of most cultivated legumes, such as pea or bean; (**B**) mutant *stamina pistilloida-1* of pea with the partial conversion of stamens into carpels; (**C**) *Vuralia turcica* with three carpels most probably emerging in successive mode; (**D**) wild-type flower of *Solanum* spp.; (**E**) flower of tomato mutants such as *fasciated* or *locule number*; (**F**); wild-type flower of most Brassicaceae; (**G**) exemplary flower of *clavata1* mutant of *Arabidopsis thaliana*; (**H**) flower of the multi-silique mutant of *Brassica napus*; (**I**) wild-type flower of rice; (**J**) flower of the *stamenless1* mutant of rice with lodicules replaced with palea/lemma-like organs and stamens with carpels (simplified). The normal gynoecium of rice is represented as unicarpellate, which is debatable. Key: dark green color = sepal/palea/lemma; white color = petal/lodicule; orange color = outer stamen; yellow color = inner stamen; light green = carpel; line between organs = fusion (not shown in a calyx and an androecium).

**Figure 2 ijms-23-09723-f002:**
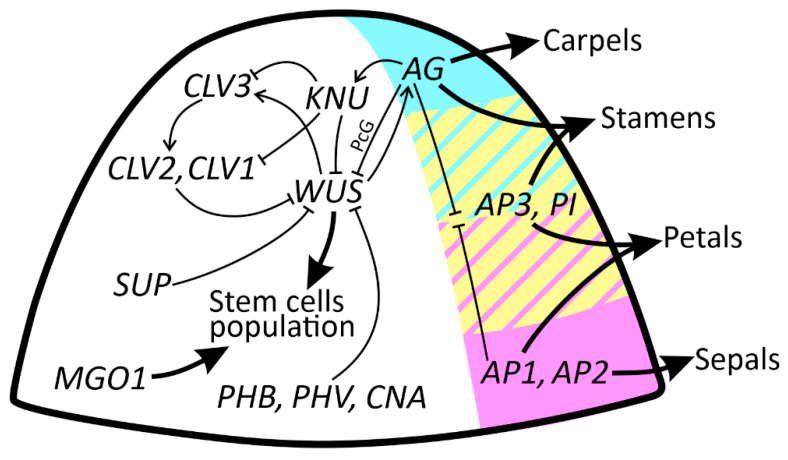
Regulatory network patterning the floral meristem and potentially affecting carpel number. Key: purple color = area of expression of A-class genes; yellow color = area of expression of B-class genes; blue color = area where C-class gene expresses; pointed arrows = positive regulation; blunt arrows = negative regulation; thicker arrows = phenotypic effect. Except for ABC genes, other genes are represented schematically without reflecting the actual sites of their expression. Gene abbreviations and detailed information on the pathways are explained in the text.

## Data Availability

Not applicable.

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
