# Peer review of "Evaluation of the Possible Contribution of Various Regulatory Genes to Determination of Carpel Number as a Potential Mechanism for Optimal Agricultural Yield"

_ijms, 2022, doi:10.3390/ijms23179723_

Round 1

Reviewer 1 Report

This is an interesting review paper about Possible Contribution of Different Regulatory Genes to Determination of Carpel Number as a Potential Mechanism for Optimal Agricultural Yield.

The study updates us with a recent overview on regulatory genes involved in carpel number determination with agronomic aspect to plant yield. The readability of the manuscript is fluent and understandable. There are some minor concerns which should be addressed before accepting for publication:

1.      Additional English editing should be done; please check the use of prepositions (I am not a native English speaker but there might be some corrections needed for the final tuning of the manuscript).

2.      The study is orientated to agricultural yield which come from seeds/frits. What about plant biomass which could be also yield? Please specify/define which kind of agricultural yield you are describing in the study according to genes involved in…

3.      Please check the style and references according to Plants journal requirements.

Reviewer 2 Report

In this literature review, Abiri et al. have started with the description of natural variations in the carpel number among plants, followed by the description of floral meristem organization and floral organ identity genes. Authors then describe the genes regulating the size and proliferation of floral meristem and how mutations in these genes can lead to multicarpelly. Finally, authors conclude by presenting some examples regarding the potential of these mutants for agriculture.

Overall, this review is timely and important considering the rise in human population and the improvement in agricultural output required to ensure global food security. I appreciate Abiri et al. for their work and wish them all the best. 

Following are some comments for authors which can be of help in improving this manuscript:

  1. Section 2 of the review titled Natural Variation in Gynoecium Structure.

This section is a bit long in my opinion. There are some places where information is repetitive (for example, lines 107-109 and lines 117-119 can be combined to make it concise).

Also, Lines 96-106 , 125-128 seem out of place considering the title of section 2 and should be integrated into section 5. 

  1. Section 4 of the review describing the contribution of floral organ identity genes to Multicarpelly.

In this section, authors describe the ABC model of flower development. But there is no discussion about its link with multicarpelly (which is the main focus of this review). In the present state, this section seems unnecessary considering the context of this review, unless authors can bring out some examples where OIGs have been shown to cause multicarpelly. 

  1. Section 6 describing AG/WUS interaction

This section needs some organization. Lines 325-328 and Lines 350-355 are describing the results from the same paper (Sun et al., 2009). This information should not be split like this and information in Lines 350-355 should be shifted with Lines 325-328. 

Lines 335-340 describe the C2H2-ZFPs. But why is this family suddenly described at this place? Authors later reveal that KNU is a C2H2-ZFP protein. I think the order should be reversed to provide continuity. Lines 335-340 should be moved after line line 344.

In Lines 364-373, authors switch to AtMIF2 and SlIMA from KNU in the previous paragraph and describe their roles in controlling FM proliferation. But then in Lines 374-381, authors again move back to the role of KNU. Authors can consider swapping positions of these paragraphs so that they can finish discussing the role of KNU before moving to other genes.

  1. Section 8. Potential of Use of Certain Mutations in Agriculture

Authors have described various examples of mutants with multicarpelly phenotype that can be exploited for higher agricultural yield. However, there is no figure showing the multicarpellary phenotype. This review would be much more engaging if the authors can include pictures of multicarpelly mutants described in section 8. Another option is to include schematic diagrams of the mutants instead of pictures. Some inspiration for such schematic can be taken from following papers:

Rapeseed: Figure 2, Xu et al., 2021 (DOI: 10.1111/pbr.12880)

Tomato: Graphical abstract, Rodríguez-Leal et al., 2017 (https://doi.org/10.1016/j.cell.2017.08.030)

With pictures or schematics, it will be much easier for the readers to visualize the increase in carpel number. 

  1. Miscellaneous comments

  • Line 325: “directly or indirectly” should be changed to “direct or indirect”.

  • Line 339: “C2H2-ZFPs are more involved in regulating flowering induction and floral organ development rather than regulating floral organ maturation.”

Authors should elaborate on what is the difference between “floral organ development” and “floral organ maturation”. Also, what do authors mean by more involved or less involved? Is there a weak phenotype or no phenotype?

  • Line 343: “containing”, “contains” words are repetitive.

  • Fig 1 legend: all the arrows in the figure are solid. Some are regular arrows, others are thicker arrows. Authors might want to use the term thicker arrows rather than solid arrows.

  • Line 346: what is the meaning of basal gynoceium structures?

  • Line 343, 357: hyphen (-) missing in C2H2-ZFPs.

  • Line 361: “negatively interacts”:  do authors mean protein protein interaction or transcriptional repression? Please modify this line to avoid confusion.

  • Fig 1: Knuckles repress CLV1 and CLV3 (Lines 374-376). Arrows for repression not shown in figure 1.

  • Line 429: Change “sized” to “size”

  • Line 458-460: “Flowers of  both pCLV3:amiR-KNU and pWUS:amiR-KNU transgenic plants generate three to four carpels compared to the wild type.” 

Three to four carpels more or three to carpels less compared to wildtype?

  • Lines 495-496: “In Arabidopsis, they transformed the Bjmc1 gene into the clv1-1 allele to over-express it.” This line is not clear.

In my opinion, a revised version of the manuscript with these changes along with the modifications suggested by other reviewers will be suitable for publication in IJMS.
